# Contextual factors and spatial trends of childhood malnutrition in Zambia

**Million Phiri[1,2]☯, David Mulemena[3]☯, Chester Kalinda[4,5]☯, Julius Nyerere Odhiambo❨iD❩[6]☯***

**1** Department of Population Studies, University of Zambia, School of Humanities and Social Sciences, Lusaka, Zambia, **2** Department of Demography and Population Studies, Schools of Public Health and Social Sciences, University of the Witwatersrand, Johannesburg, South Africa, **3** Ministry of Mines and Minerals Development, Lusaka, Zambia, **4** University of Global Health Equity (UGHE), Bill and Joyce Cummings Institute of Global Health, Institute of Global Health Equity Research (IGHER), Kigali, Rwanda, **5** Institute of Global Health Equity Research (IGHER), University of Global Health Equity, Kigali, Rwanda, **6** Ignite Global Health Research Lab, Global Research Institute, William and Mary, Williamsburg, Virginia, United States of America

☯ These authors contributed equally to this work.
* nyererejulius7@gmail.com

## Abstract

### Background

Understanding the national burden and epidemiological profile of childhood malnutrition is central to achieving both national and global health priorities. However, national estimates of malnutrition often conceal large geographical disparities. This study examined the prevalence of childhood malnutrition across provinces in Zambia, changes over time, and identified factors associated with the changes.

### Methods

We analyzed data from the 2013/4 and 2018 Zambia demographic and health surveys (ZDHS) to examine the spatial heterogeneity and mesoscale correlates of the dual burden of malnutrition in children in Zambia. Maps illustrating the provincial variation of childhood malnutrition were constructed. Socio-demographic and clinical factors associated with childhood malnutrition in 2013 and 2018 were assessed independently using a multivariate logistic model.

### Results

Between 2013/4 and 2018, the average prevalence of stunting decreased from 40.1% (95% CI: 39.2–40.9) to 34.6% (95% CI:33.6–35.5), wasting decreased from 6.0% (95% CI: 5.6–6.5) to 4.2% (95% CI: 3.8–4.7), underweight decreased from 14.8% (95% CI: 14.1–15.4) to 11.8% (95% CI: 11.2–12.5) and overweight decreased from 5.7% (95% CI: 5.3–6.2) to 5.2% (95% CI: 4.8–5.7). High variability in the prevalence of childhood malnutrition across the provinces were observed. Specifically, stunting and underweight in Northern and Luapula provinces were observed in 2013/14, whereas Lusaka province had a higher degree of variability over the two survey periods.

**Data Availability Statement:** The data underlying the results presented in the study are available from the Demographic Health Survey website at https://dhsprogram.com/data/.

**Funding:** The author(s) received no specific funding for this work.

**Competing interests:** The authors have declared that no competing interests exist.

**Abbreviations:** AOR, Adjusted odds ratio; OR, Odds ratio; CI, Confidence Interval; CPH, Census of Population and Housing; DHS, Demographic and Health Survey; HIV, Human Immuno-deficiency virus; ZDHS, Zambia Demographic and Health Survey; ZMPR, Zambia Population Recode File; WHO, World Health Organization; UNICEF, United Nations Children's Fund.

## Conclusion

The study points to key sub-populations at greater risk and provinces where malnutrition was prevalent in Zambia. Overall, these results have important implications for nutrition policy and program efforts to reduce the double burden of malnutrition in Zambia.

## Introduction

Malnutrition refers to excesses, deficiencies, or imbalances in a person's intake of nutrients and energy. In children, it remains a persistent global public health problem that accounts for half of the global childhood mortality and impairs childhood physical and cognitive development [1, 2]. It endangers a child's success as an adult by reducing their productivity and making them more susceptible to premature death. Studies suggest that reversing the negative effects of malnutrition on cognitive development sorely through nutrition-based interventions is ineffective unless integrated with medical treatment or social enrichment [2, 3]. To reduce the detrimental effects of malnutrition on childhood development, the United Nations Decade of Action on Nutrition (2016–2025) has been developed to provide a unique and time-bound opportunity to address the burden of malnutrition worldwide [4]. Halfway through the implementation period, ending malnutrition appears insufficient and limited, especially in sub-Saharan Africa where infectious diseases, health inequalities, and food insecurity remain unresolved [5]. Furthermore, the dual burden of poverty and malnutrition in this region increases the difficulties associated with obtaining data on the geographical distribution and hotspots of childhood overnutrition [6, 7].

Various intervention strategies and programs have been launched to reduce malnutrition [8, 9]. The intervention strategies have been developed on the premise of achieving one of the Sustainable Development Goals (SDGs) Target 2.2 which focuses on ending all forms of malnutrition. The success of these activities may depend on scaling up access to safe and effective health interventions, quantifying the nutritional status by mapping changes, and identifying hot spots of malnutrition to give insights into the spread of the problem. In Zambia, nutrition intervention programs to address different forms of malnutrition using integrated approaches have been implemented [10, 11]. Yet, malnutrition persists and the emerging dual burden of malnutrition [12] introduces additional obstacles to the realization of the Zambia National Health Strategic Plan (ZNHSP) of 201–21 whose overarching nutrition objective is to "reduce under and over nutrition and improve clinical nutrition by 2021" [13]. Therefore, determining the spatial epidemiology of childhood under and overnutrition and exploring associated characteristics would be vital in developing and focusing the public health policy on redesigning effective malnutrition intervention and prevention approaches.

A vital component of spatial epidemiology lies in its ability to provide an articulate visual summary of the spatial phenomena that may be challenging to achieve using conventional tabular representations. Previous studies quantifying childhood malnutrition in Zambia have used hospital-based data [14, 15]. However, the burden may vary geographically thus, its detection in time and space allows focused and tailored intervention strategies. Here, the study also provides the basis for mounting meaningful suites of nutrition-based interventions at the provincial level and gives a glimpse of the temporal progress and the contribution of socio-economic determinants, communicable and non-communicable diseases on the epidemiological transition of childhood malnutrition.

## Methods

### Study design

This study is based on a secondary analysis of the existing data from the International Demographic and Health Survey (DHS) programme. A detailed description of the methods used in these surveys is included in the survey reports for Zambia [16, 17]. The current study utilized data from the 2013–14 and 2018 Zambia Demographic and Health Surveys (ZDHS). The nature of DHS data allowed for comparisons between and over periods. This allows the monitoring of changes in key indicators of variables of interest in different geographical areas. The data analyzed in this paper relate under and over-nutrition among children aged under five in households systematically selected from a household listing of all households in the enumeration area. To determine the anthropometric measures, children between 0–59 months from households who consented, were enrolled in the study. The women respondents who participated in the study were aged between 15–49 years. The 2013–14 and 2018 DHS captured samples of 12,328 and 9,689 children aged between 0–59 months, respectively.

### Outcome measures

The outcome variable of interest in this study was childhood malnutrition (stunting, wasting, underweight, and overweight). The WHO standards were used to define the anthropometric indicators of nutritional status [18]. Relative to WHO standards, children with a Z score less than minus 2 (-2) were classified as undernourished while those with a Z score greater than 2 (+2). Weight-for-age was used to define underweight; height-for-age for stunting; and weight-for-height for wasting. Although each indicator shows different nutritional statuses, deviations below—2 standard deviations (SD) show moderate or severe undernourishment among children. As previously defined, we considered malnutrition to include both under and over-nutrition [7, 19]. Using the published WHO conceptual framework on the determinant of childhood malnutrition [20] and 2018 WHO global nutrition report [21], we identified factors that could be potentially associated with malnutrition among children aged under five. The two datasets, DHS reference materials, and data collection forms were used to identify key variables. These variables were classified at three levels: individual level, household level, and maternal level. The datasets of interest were limited to community-level information hence all analysis was based on the provincial level for which the data had been aggregated.

### Statistical analysis

Data analyses were performed using Stata version 15 (Stata Corp 2015, College Station, TX) taking into consideration survey design, cluster effect, and post-stratification weights. Key socio-economic and demographic factors were described and expressed in frequency and percentage. The significant differences were assessed between the two survey periods using the Chi-square test. Exploratory bivariate analysis was carried out separately for two datasets to assess the association between the prevalence of childhood malnutrition and selected variables (child's age in months, sex of child, residence, mother's education level, household wealth index, child's birth weight, birth interval, mother employment status, water source and presence of diarrhea).

To assess the progressive health in the various groups, age was categorized into classes. Variables that were observed to have a statistically significant association with our indicators were included in the multivariate analysis. Four multivariate logistic regression models were then used to assess the independent factors associated with childhood malnutrition outcomes. Odds ratios with their corresponding 95% confidence intervals after adjusting for all our independent variables were reported. The initial logistic models included all significantly

associated factors from the bivariate analyses as well as factors that had a *p*-value of <0.20. Also, interested covariates were included in the multivariate analyses regardless of their significant levels. The prevalence of stunting, wasting, underweight and overweight and its distribution across 10 provinces was estimated. The results were then used to construct maps highlighting the provincial variation in prevalence and trends between the two survey periods using ArcMap 10.7.1 (ESRI Inc., Redlands, CA, USA).

## Ethical considerations

The 2013–14 and 2018 Zambia Demographic and Health Survey protocols for survey methodology and biomarker measurements were approved by both the Inner City Fund (ICF) institutional review boards (IRBs) and the Tropical Diseases Research Centre (TDRC) in Zambia. Both IRBs approved the protocols before the commencement of data collection activities. Study protocols were carried out following relevant guidelines and regulations on confidentiality, benevolence, non-maleficence, and informed consent. The study participants gave written informed consent before participation and all information was collected confidentially. Permission to use DHS data for this study was sought from ICF international [27].

## Results

### Bivariate analysis of childhood malnutrition prevalence in 2013 and 2018 by background characteristics

Overall, 20, 588 anthropometric indices (12,327 and 8,261 for 2013/14 and 2018, respectively) for children aged between 0–59 months were available in the two DHS waves under study. Approximately 66.1% and 65.6% of the children were from rural populations in 2013 and 2018, respectively. In both periods, the prevalence of malnutrition across all indicators, was substantially higher in children born in poor households, although this association was only significant for children who had stunted growth and were underweight (Table 1). The prevalence of stunting, and underweight among children born to mothers with no schooling, was significantly higher when compared to children born to mothers with primary, secondary, and tertiary education. However, the prevalence of overweight was higher for children born to women with tertiary education (8.8%). The presence of diarrhea was significantly associated with stunting, wasting, and underweight in children. There was no significant association between diarrhea and overweight reported in both 2013/14 and 2018.

Between 2013 and 2018, the average prevalence of stunting decreased from 40.1% (95% CI: 39.2–40.9) to 34.6% (95% CI:33.6–35.5), wasting decreased from 6.0% (95% CI: 5.6–6.5) to 4.2% (95% CI: 3.8–4.7), underweight decreased from 14.8% (95% CI: 14.1–15.4) to 11.8% (95% CI: 11.2–12.5) and overweight decreased from 5.7% (95% CI: 5.3–6.2) to 5.2% (95% CI: 4.8–5.7). Fig 1 shows the prevalence rates of malnutrition indicators across all age groups, wealth index, and maternal education. Stunting was strongly associated with age (p < 0.01) in both DHS waves. Both in 2014 and 2018, stunting and underweight were most prevalent in children aged 18–23 months. Furthermore, these two indicators were also prevalent in children coming from the poorest households and those whose mothers had no formal education. In both waves, an increase in the prevalence of overweight children was observed in both waves among those from the richest households and those whose mothers had attained tertiary education.

### Analysis of malnutrition prevalence by province

The results in Fig 2 show the standard deviation maps of malnutrition in Zambia. The maps show higher uncertainties in the prevalence of childhood malnutrition across different

**Table 1. Bivariate analysis of childhood malnutrition indicators with background characteristics (2013/14 DHS data).**

| Background Characteristics | 2013–14 DHS | | | | | | | | | | | |
|---|---|---|---|---|---|---|---|---|---|---|---|---|
| | Stunted | No-stunted | p-value | Wasted | No-wasted | p-value | Underweight | No-underweight | p-value | Overweight | No-overweight | p-value |
| | N = 12,328 | | | N = 12,328 | | | N = 12,328 | | | N = 12,328 | | |
| **Age in months** | | | | | | | | | | | | |
| < 6 months | 13.6 | 86.4 | 0.000 | 8 | 92 | 0.000 | 5.8 | 94.2 | 0.000 | 15.4 | 84.6 | 0.000 |
| 6–8 | 25.1 | 74.9 | | 9 | 91 | | 11.4 | 88.6 | | 11.6 | 88.4 | |
| 9–11 | 38.5 | 61.5 | | 10.1 | 89.9 | | 17.4 | 82.6 | | 10.4 | 89.6 | |
| 12–17 | 43.1 | 56.9 | | 7.6 | 92.4 | | 13.5 | 86.5 | | 6.1 | 93.9 | |
| 18–23 | 54 | 46 | | 6.1 | 93.9 | | 17.8 | 82.2 | | 5.7 | 94.3 | |
| 24–35 | 51 | 49 | | 5.2 | 94.8 | | 17.1 | 82.9 | | 4 | 96 | |
| 36–47 | 41.6 | 58.4 | | 5.1 | 94.9 | | 15.2 | 84.8 | | 3.4 | 96.6 | |
| 48–59 | 34.6 | 65.4 | | 4.5 | 95.5 | | 14.9 | 85.1 | | 3.3 | 96.7 | |
| **Sex of Child** | | | | | | | | | | | | |
| Male | 42.4 | 57.6 | 0.000 | 6.2 | 93.8 | 0.440 | 16 | 84 | 0.001 | 6.1 | 93.9 | 0.167 |
| Female | 37.6 | 62.4 | | 5.8 | 94.2 | | 13.5 | 86.5 | | 5.4 | 94.6 | |
| **Residence** | | | | | 100 | | | | | | 100 | |
| Urban | 36 | 64 | 0.000 | 6.4 | 93.6 | 0.399 | 12.9 | 87.1 | 0.004 | 6.5 | 93.5 | 0.066 |
| Rural | 42.1 | 57.9 | | 5.9 | 94.1 | | 15.7 | 84.3 | | 5.4 | 94.6 | |
| **Size at birth** | | | | | | | | | | | | |
| Very small | 62.1 | 37.9 | 0.000 | 9.3 | 90.7 | 0.001 | 32.4 | 67.6 | 0.000 | 2.7 | 97.3 | 0.127 |
| Small | 51.5 | 48.5 | | 9.2 | 90.8 | | 25.8 | 74.2 | | 7.1 | 92.9 | |
| Average or larger | 38.3 | 61.7 | | 5.7 | 94.3 | | 13.2 | 86.8 | | 8.3 | 91.7 | |
| **Birth Interval** | | | | | | | | | | | | |
| First birth | 40 | 60 | 0.000 | 6 | 94 | 0.068 | 14.5 | 85.5 | 0.000 | 5.8 | 94.2 | 0.403 |
| Less than 24 months | 46.1 | 53.9 | | 5.8 | 94.2 | | 19.6 | 80.4 | | 5.7 | 94.3 | |
| 24–47 months | 40.5 | 59.5 | | 5.6 | 94.4 | | 14.7 | 85.3 | | 5.6 | 94.4 | |
| 48+ months | 34.4 | 65.6 | | 7.5 | 92.5 | | 12.2 | 87.8 | | 6.6 | 93.4 | |
| **Wealth index** | | | | | | | | | | | | |
| Poorest | 47.3 | 52.7 | 0.000 | 6.8 | 93.2 | 0.387 | 20.1 | 79.9 | 0.000 | 5.3 | 94.7 | 0.237 |
| Poor | 41.7 | 58.3 | | 5.5 | 94.5 | | 15.7 | 84.3 | | 4.9 | 95.1 | |
| Middle | 40.7 | 59.3 | | 4.9 | 95.1 | | 13.7 | 86.3 | | 6.2 | 93.8 | |
| Richer | 37.6 | 62.4 | | 4.4 | 95.6 | | 12.7 | 87.3 | | 6.1 | 93.9 | |
| Richest | 28.4 | 71.6 | | 4 | 96 | | 8.9 | 91.1 | | 6.7 | 93.3 | |
| **Mother's Education** | | | | | | | | | | | | |
| None | 44.4 | 55.6 | 0.000 | 7.1 | 92.9 | 0.504 | 20 | 80 | 0.000 | 6 | 94 | 0.113 |
| Primary | 41.9 | 58.1 | | 5.8 | 94.2 | | 15.5 | 84.5 | | 5.6 | 94.4 | |
| Secondary | 36.9 | 63.1 | | 6.1 | 93.9 | | 12 | 88 | | 5.9 | 94.1 | |
| Tertiary | 17.9 | 82.1 | | 5.4 | 94.6 | | 4.8 | 95.2 | | 8.8 | 91.2 | |
| **Mother's occupation status** | | | | | | | | | | | | |
| Not employed | 39.5 | 60.5 | 0.5696 | 5.8 | 94.2 | 0.411 | 13.7 | 86.3 | 0.086 | 6 | 94 | 0.808 |
| Employed | 40.2 | 59.8 | | 6.2 | 93.8 | | 15.3 | 84.7 | | 5.8 | 94.2 | |
| **Region** | | | | | | | | | | | | |

*(Continued)*

**Table 1.** (Continued)

| Background Characteristics | 2013–14 DHS | | | | | | | | | | | |
|---|---|---|---|---|---|---|---|---|---|---|---|---|
| | Stunted | No-stunted | p-value | Wasted | No-wasted | p-value | Underweight | No-underweight | p-value | Overweight | No-overweight | p-value |
| | N = 12,328 | | | N = 12,328 | | | N = 12,328 | | | N = 12,328 | | |
| Central | 42.5 | 57.5 | 0.000 | 4.6 | 95.4 | 0.000 | 15.3 | 84.7 | 0.000 | 6.7 | 93.3 | 0.002 |
| Copperbelt | 36.2 | 63.8 | | 5.8 | 94.2 | | 14.1 | 85.9 | | 5.2 | 94.8 | |
| Eastern | 43.3 | 56.7 | | 5.1 | 94.9 | | 12.8 | 87.2 | | 6 | 94 | |
| Luapula | 43 | 57 | | 13.1 | 86.9 | | 21.2 | 78.8 | | 4.9 | 95.1 | |
| Lusaka | 35.7 | 64.3 | | 7 | 93 | | 11 | 89 | | 8 | 92 | |
| Muchinga | 43.6 | 56.4 | | 4.1 | 95.9 | | 15.6 | 84.4 | | 5.2 | 94.8 | |
| Northern | 58.5 | 41.5 | | 3.7 | 96.3 | | 19 | 81 | | 5.3 | 94.7 | |
| North western | 36.9 | 63.1 | | 8.2 | 91.8 | | 13.8 | 86.2 | | 7.4 | 92.6 | |
| Southern | 37.2 | 62.8 | | 4.2 | 95.8 | | 13.1 | 86.9 | | 4.6 | 95.4 | |
| Western | 36.2 | 63.8 | | 6.5 | 93.5 | | 16.2 | 83.8 | | 3.1 | 96.9 | |
| **Source of water** | | | | | | | | | | | | |
| Improved | 37.4 | 62.6 | 0.000 | 5.9 | 94.1 | 0.433 | 13.1 | 86.9 | 0.000 | 6 | 94 | 0.195 |
| Non improved | 43.8 | 56.2 | | 6.3 | 93.7 | | 17.1 | 82.9 | | 5.3 | 94.7 | |
| **Presence of Diarrhea** | | | | | | | | | | | | |
| No | 38.1 | 61.9 | 0.001 | 5.8 | 94.2 | 0.036 | 14 | 86 | 0.001 | 6 | 94 | 0.186 |
| Yes, in last 2 weeks | 43.8 | 56.2 | | 7.3 | 92.7 | | 17.5 | 82.5 | | 5.2 | 94.8 | |
| **Total** | **40.1** | **59.9** | | **6** | **94** | | **14.8** | **85.2** | | **5.7** | **94.3** | |

provinces in Zambia. Over the two study periods, stunting in Northern province and Luapula provinces was highly uncertain when compared to other provinces. In 2013/14 wasting was highly uncertain in Luapula. For underweight, Luapula and Northern provinces had the highest degree of uncertainty in 2013/14. This trend was consistent in 2018 with Luapula, Northern, Muchinga, and Western provinces having the highest levels of uncertainty. For overweight, Lusaka province had a higher degree of uncertainty over the two survey periods. However, in 2018, the Northern province also exhibited a higher degree of uncertainty.

## Multivariable analysis results and the association with socioeconomic determinants

**Stunting.** Results of the multivariable logistic regression of malnutrition indicators and its associated explanatory variables showed that compared to those who were not stunted, the unadjusted odds of stunting in 2014/13 and 2018 for children in the age group 18–23 were 7.81 (95%CI: 6.02–10.12) and 3.99 (95% CI: 3.06–5.19) times respectively, higher than those aged less than 6 months. Furthermore, an association between gender and stunting was observed with female children when compared to their male counterparts were 22% (0.78; 95% CI: 0.72–0.87) less likely to be stunted in 2013/4 and 32% (0.68; 95% CI: 0.61–0.76) less likely to be stunted compared in 2018. In 2013/4 and 2018, children from mothers with tertiary education were 57% (0.43; 95% CI: 0.27–6.02) and 55% (0.45; 95%CI: 0.28–0.73) less likely to be stunted, respectively. Both in 2013/14 and 2018, children from mothers who had a parity interval of more than 48 months were 16% (0.84; 95%CI: 0.71–0.99) and 17% (0.83; 95% CI: 0.71–0.97) less likely to be stunted when compared to children whose mother had the first birth (Table 2).

**Wasting.** Our results show that the odds of wasting reduced with age. In 2013/14, children aged between 48–59 months old were 44% (0.56; 95% CI: 0.39–0.79) less likely to be wasted

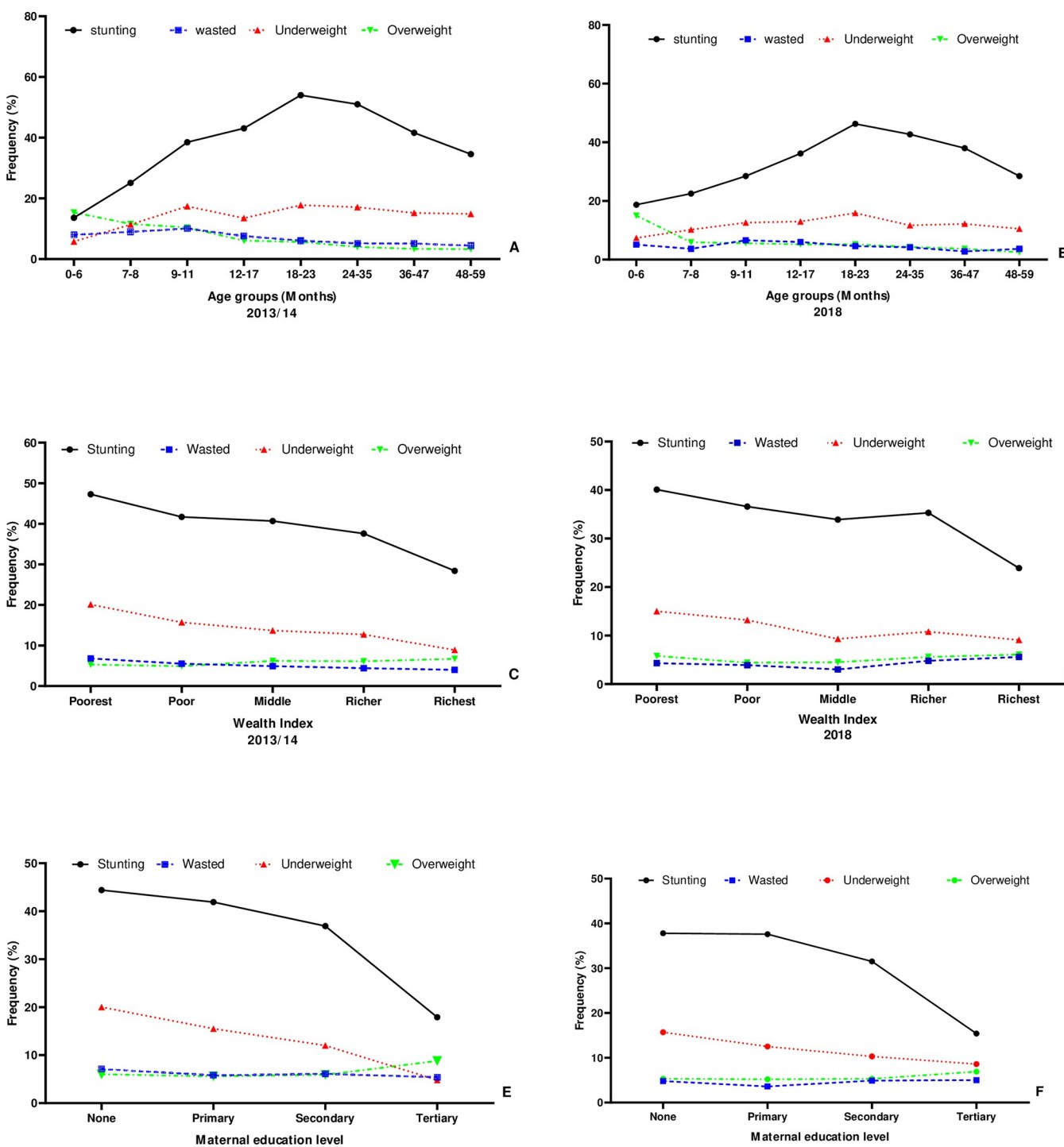

**Fig 1.** Trends of malnutrition Indicators by (A) Age in 2013/14 (B) Age in 2018 (C) Wealth Index 2013/14 (D) Wealth Index 2018 (E) Maternal education level in 2013/14 (F) Maternal education level in 2018. (Source: author generated map).

compared to those aged less than 6 months. Our results from the model further indicate that maternal characteristics such as the mother's education level, parity, and mother's employment status had no significant association with wasting. However, in 2013/14 and 2018,

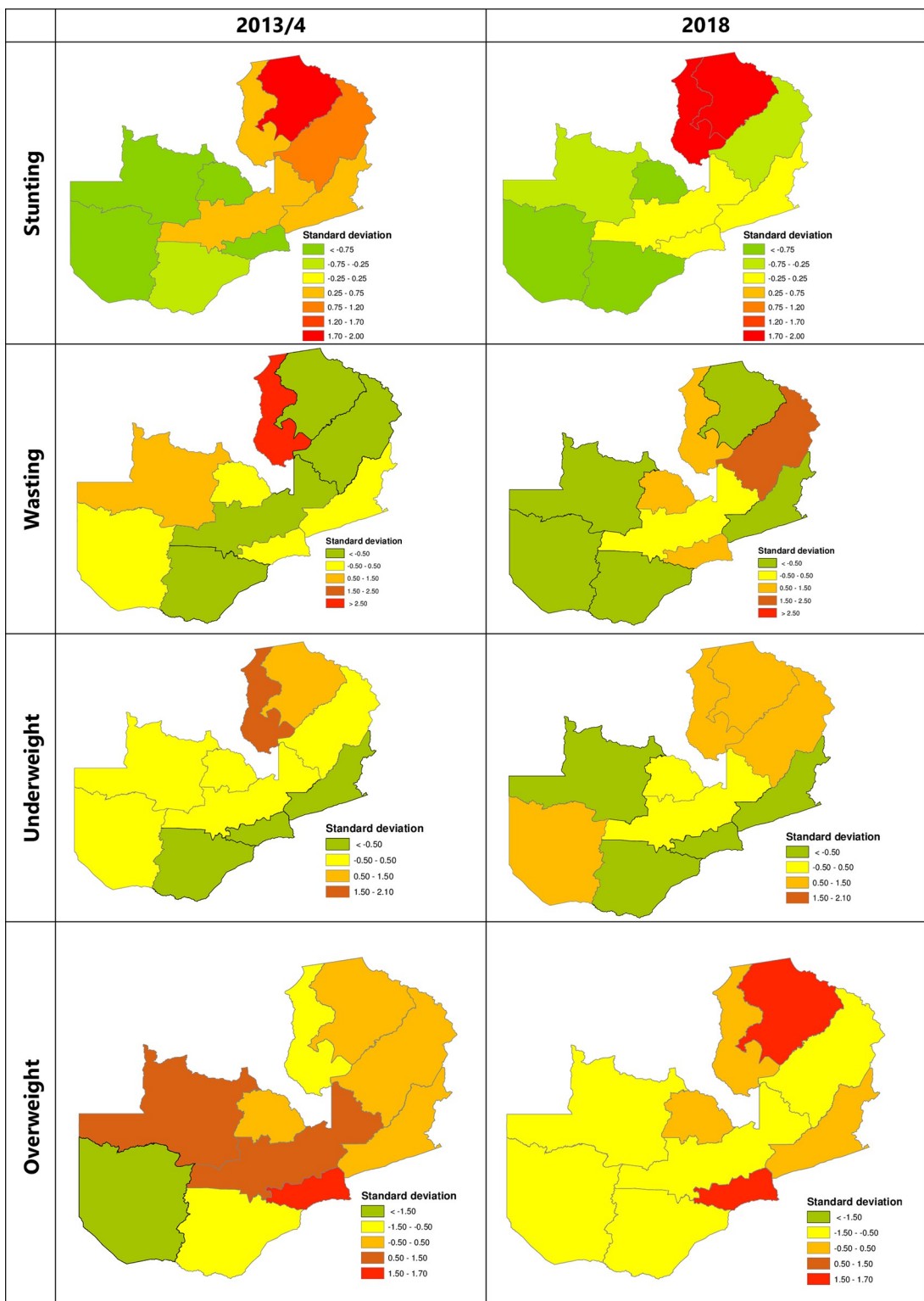

**Fig 2. Spatial variation of stunting prevalence, wasting prevalence, and underweight prevalence by provinces in Zambia.**
Generated with ArcMap 10.7 by ESRI (https://desktop.arcgis.com/en /). (Source: author generated map).

**Table 2. Bivariate analysis of childhood malnutrition indicators with background characteristics (2018 DHS data).**

| Background Characteristics | 2018 DHS | | | | | | | | | | | |
|---|---|---|---|---|---|---|---|---|---|---|---|---|
| | Stunted | No-stunted | p-value | Wasted | No-wasted | p-value | Underweight | No-underweight | p-value | Overweight | No-overweight | p-value |
| | N = 12,328 | | | N = 12,328 | | | N = 12,328 | | | N = 12,328 | | |
| **Age in months** | | | | | | 0.021 | | | 0.001 | | | |
| < 6 months | 18.7 | 81.3 | 0.000 | 5.1 | 94.9 | | 7.4 | 92.6 | | 15 | 85 | 0.000 |
| 6–8 | 22.5 | 77.5 | | 3.7 | 96.3 | | 10.2 | 89.8 | | 5.9 | 94.1 | |
| 9–11 | 28.5 | 71.5 | | 6.6 | 93.4 | | 12.6 | 87.4 | | 5.6 | 94.4 | |
| 12–17 | 36.2 | 63.8 | | 6 | 94 | | 13 | 87 | | 5.2 | 94.8 | |
| 18–23 | 46.3 | 53.7 | | 4.6 | 95.4 | | 15.9 | 84.1 | | 5.2 | 94.8 | |
| 24–35 | 42.7 | 57.3 | | 4.2 | 95.8 | | 11.7 | 88.3 | | 4.3 | 95.7 | |
| 36–47 | 38.0 | 62.0 | | 2.8 | 97.2 | | 12.2 | 87.8 | | 3.7 | 96.3 | |
| 48–59 | 28.5 | 71.5 | | 3.7 | 96.3 | | 10.5 | 89.5 | | 2.5 | 97.5 | |
| **Sex of Child** | | | | | | | | | | | | |
| Male | 38.3 | 61.7 | 0.000 | 4.8 | 95.2 | 0.046 | 13.5 | 86.5 | 0.000 | 5.4 | 94.6 | 0.447 |
| Female | 31.0 | 69.0 | | 3.7 | 96.3 | | 10.2 | 89.8 | | 5 | 95 | |
| **Residence** | | | | | | | | | | | | |
| Urban | 32.1 | 67.9 | 0.006 | 5 | 95 | 0.091 | 10.8 | 89.2 | 0.129 | 5.7 | 94.3 | 0.292 |
| Rural | 35.9 | 64.1 | | 3.8 | 96.2 | | 12.4 | 87.6 | | 5 | 95 | |
| **Size at birth** | | | | | | | | | | | | |
| Very small | 49.9 | 50.1 | 0.000 | 6.1 | 93.9 | 0.023 | 24.1 | 75.9 | 0.000 | 3.6 | 96.4 | 0.202 |
| Small | 46.5 | 53.5 | | 6 | 94 | | 21.5 | 78.5 | | 3.7 | 96.3 | |
| Average or larger | 33.0 | 67.0 | | 3.9 | 96.1 | | 10.3 | 89.7 | | 5.5 | 94.5 | |
| **Birth Interval** | | | | | | | | | | | | |
| First birth | 35.0 | 65.0 | 0.000 | 4.4 | 95.6 | 0.076 | 12.5 | 87.5 | 0.000 | 4.9 | 95.1 | 0.092 |
| Less than 24 months | 42.1 | 57.9 | | 4.1 | 95.9 | | 17 | 83 | | 4.9 | 95.1 | |
| 24–47 months | 35.1 | 64.9 | | 3.5 | 96.5 | | 10.7 | 89.3 | | 4.8 | 95.2 | |
| 48+ months | 30.5 | 69.5 | | 5.3 | 94.7 | | 10.5 | 89.5 | | 6.5 | 93.5 | |
| **Wealth index** | | | | | | | | | | | | |
| Poorest | 40.1 | 59.9 | 0.000 | 4.3 | 95.7 | 0.090 | 15 | 85 | 0.000 | 5.8 | 94.2 | 0.207 |
| Poor | 36.6 | 63.4 | | 3.9 | 96.1 | | 13.2 | 86.8 | | 4.4 | 95.6 | |
| Middle | 33.9 | 66.1 | | 3 | 97 | | 9.3 | 90.7 | | 4.5 | 95.5 | |
| Richer | 35.3 | 64.7 | | 4.8 | 95.2 | | 10.8 | 89.2 | | 5.6 | 94.4 | |
| Richest | 23.9 | 76.1 | | 5.6 | 94.4 | | 9.1 | 90.9 | | 6.1 | 93.9 | |
| **Mother's Education** | | | | | | | | | | | | |
| None | 37.8 | 62.2 | 0.000 | 4.8 | 95.2 | 0.183 | 15.7 | 84.3 | 0.004 | 5.3 | 94.7 | 0.715 |
| Primary | 37.6 | 62.4 | | 3.6 | 96.4 | | 12.5 | 87.5 | | 5.2 | 94.8 | |
| Secondary | 31.5 | 68.5 | | 4.9 | 95.1 | | 10.3 | 89.7 | | 5.3 | 94.7 | |
| Tertiary | 15.4 | 84.6 | | 5 | 95 | | 8.6 | 91.4 | | 6.9 | 93.1 | |
| **Mother's occupation status** | | | | | | | | | | | | |
| Not employed | 34.5 | 65.5 | 0.991 | 4.7 | 95.3 | 0.203 | 12.9 | 87.1 | 0.020 | 6 | 94 | 0.029 |
| Employed | 34.5 | 65.5 | | 3.8 | 96.2 | | 10.8 | 89.2 | | 4.6 | 95.4 | |
| **Region** | | | | | | | | | | | | |

(*Continued*)

**Table 2.** (Continued)

| Background Characteristics | 2018 DHS | | | | | | | | | | | |
|---|---|---|---|---|---|---|---|---|---|---|---|---|
| | Stunted | No-stunted | p-value | Wasted | No-wasted | p-value | Underweight | No-underweight | p-value | Overweight | No-overweight | p-value |
| | N = 12,328 | | | N = 12,328 | | | N = 12,328 | | | N = 12,328 | | |
| Central | 33.4 | 66.6 | 0.000 | 4.6 | 95.4 | 0.000 | 11.4 | 88.6 | 0.010 | 3.9 | 96.1 | 0.000 |
| Copperbelt | 29.7 | 70.3 | | 5.4 | 94.6 | | 12.1 | 87.9 | | 5 | 95 | |
| Eastern | 34.2 | 65.8 | | 2.2 | 97.8 | | 9.2 | 90.8 | | 5 | 95 | |
| Luapula | 44.9 | 55.1 | | 6.2 | 93.8 | | 15.2 | 84.8 | | 5.2 | 94.8 | |
| Lusaka | 35.6 | 64.4 | | 5.5 | 94.5 | | 10.6 | 89.4 | | 8.1 | 91.9 | |
| Muchinga | 32.1 | 67.9 | | 8.2 | 91.8 | | 15.3 | 84.7 | | 3.5 | 96.5 | |
| Northern | 45.8 | 54.2 | | 3.1 | 96.9 | | 14.1 | 85.9 | | 8.3 | 91.7 | |
| North western | 31.9 | 68.1 | | 2.4 | 97.6 | | 10.4 | 89.6 | | 3.3 | 96.7 | |
| Southern | 29.4 | 70.6 | | 2.3 | 97.7 | | 9.7 | 90.3 | | 3.8 | 96.2 | |
| Western | 29.0 | 71.0 | | 3 | 97 | | 14.1 | 85.9 | | 3 | 97 | |
| **Source of water** | | | | | | | | | | | | |
| Improved | 32.9 | 67.1 | 0.000 | 4.4 | 95.6 | 0.369 | 11 | 89 | 0.003 | 5.5 | 94.5 | 0.287 |
| Non improved | 38.0 | 62.0 | | 3.9 | 96.1 | | 13.5 | 86.5 | | 4.8 | 95.2 | |
| **Presence of Diarrhea** | | | | | | | | | | | | |
| No | 34.5 | 65.5 | 0.627 | 4 | 96 | 0.114 | 11.1 | 88.9 | 0.000 | 5.4 | 94.6 | 0.100 |
| Yes, in last 2 weeks | 35.3 | 64.7 | | 5.3 | 94.7 | | 15.3 | 84.7 | | 4.2 | 95.8 | |
| **Total** | **34.6** | **65.4** | | **4.2** | **95.8** | | **11.8** | **88.2** | | **5.2** | **94.8** | |

children from middle-income households were 32% (0.68 95% CI: 0.50–0.94) and 41% (0.59, 95% CI: 0.38–0.94) less likely to be wasted when compared to those from poorest households (Table 3).

**Underweight.** In 2013/14 and 2018, children aged 18–23 months were 3.32 (95% CI: 2.35–4.70) and 2.17 (95% CI: 1.49–3.17) times more likely to be underweight when compared to those aged less than 6 months. Children from mothers with tertiary education were 64% (0.36, 95% CI: 0.19–0.67) and 15% (0.85, 95% CI: 0.45–1.60) less likely to be underweight in 2013/4 and 2018 respectively. Children from the richest families were 67% (0.33, 95% CI: 0.23–0.46) and 44% (0.56, 95% CI: 0.35–0.88) less likely to be underweight compared to those from poor households in 2013/14 and 2018 respectively. Furthermore, children who at birth were classified as average or large were 66% (0.34, 95% CI: 0.23–0.50) less likely to be underweight compared to those who were classified as small. In 2018, children with diarrhea were 1.24 (95% CI: 1.02–1.50) times more likely to be underweight compared to those who did have diarrhea (Table 4).

**Overweight.** The results showed that the odds of being overweight in 2013/14 and 2018 reduced with age. Children aged 48–59 months were 83% (0.17, 95% CI: 0.12–0.23) and 86% (0.14; 95% CI: 0.09–0.21) less likely to be overweight in 2013/4 and 2018 respectively, when compared to those below 6 months. Furthermore, in 2013/4, children with average or large birth weight were 2.32 (95% CI: 1.01–5.32) times more likely to be overweight compared to those classified as small at birth. None of the household characteristics and maternal characteristics were observed to have a significant relationship with overweight in both 2013/14 and 2018 (Table 4).

## Discussion

The study describes variations in the prevalence of childhood malnutrition (stunting, wasting, underweight, overweight) in Zambia using selected individual, household, and maternal

**Table 3. Multivariate analysis of malnutrition indicators (stunting and wasting) for 2013/14 and 2018 DHS waves.**

| | Stunting | | | Wasting | | | Stunting | | | Wasting | | |
|---|---|---|---|---|---|---|---|---|---|---|---|---|
| | AOR | p-values | [95% CI] | AOR | p-values | [95% CI] | AOR | p-values | [95% CI] | AOR | p-values | [95% CI] |
| **Child's Age in months** | | | | | | | | | | | | |
| Less than 6 months | 1 | | | 1 | | | 1 | | | 1 | | |
| 6–8 | 2.100 | 0.000 | (1.528 2.887) | 1.153 | 0.549 | (0.724 1.834) | 1.340 | 0.080 | (0.965 1.861) | 0.716 | 0.443 | (0.304 1.695) |
| 9–11 | 4.128 | 0.000 | (3.012 5.659) | 1.203 | 0.403 | (0.781 1.850) | 1.704 | 0.001 | (1.229 2.362) | 1.374 | 0.349 | (0.706 2.672) |
| 12–17 | 5.094 | 0.000 | (3.946 6.576) | 0.897 | 0.557 | (0.623 1.291) | 2.836 | 0.000 | (2.140 3.758) | 1.226 | 0.498 | (0.679 2.213) |
| 18–23 | 7.805 | 0.000 | (6.022 10.116) | 0.724 | 0.08 | (0.505 1.039) | 3.988 | 0.000 | (3.066 5.185) | 0.909 | 0.765 | (0.486 1.701) |
| 24–35 | 7.045 | 0.000 | (5.534 8.912) | 0.615 | 0.006 | (0.434 0.871) | 3.447 | 0.000 | (2.753 4.315) | 0.816 | 0.408 | (0.503 1.323) |
| 36–47 | 4.699 | 0.000 | (3.684 5.993) | 0.605 | 0.003 | (0.432 0.846) | 2.919 | 0.000 | (2.321 3.671) | 0.610 | 0.080 | (0.351 1.061) |
| 48–59 | 3.387 | 0.000 | (2.684 4.273) | 0.561 | 0.001 | (0.397 0.793) | 1.802 | 0.000 | (1.409 2.306) | 0.803 | 0.356 | (0.503 1.281) |
| **Sex of Child** | | | | | | | | | | | | |
| Male | 1 | | | 1 | | | 1 | | | 1 | | |
| Female | 0.789 | 0.000 | (0.719 0.865) | 0.907 | 0.289 | (0.756 1.087) | 0.678 | 0.000 | (0.608 0.755) | 0.782 | 0.090 | (0.589 1.039) |
| **Residence** | | | | | | | | | | | | |
| Urban | 1 | | | 1 | | | 1 | | | 1 | | |
| Rural | 0.795 | 0.004 | (0.680 0.928) | 0.743 | 0.07 | (0.538 1.025) | 0.770 | 0.009 | (0.633 0.937) | 0.891 | 0.564 | (0.601 1.320) |
| **Education level of mother** | | | | | | | | | | | | |
| No education | 1 | | | 1 | | | 1 | | | 1 | | |
| Primary | 0.930 | 0.405 | (0.784 1.103) | 0.842 | 0.232 | (0.635 1.116) | 1.062 | 0.535 | (0.878 1.284) | 0.838 | 0.459 | (0.525 1.339) |
| Secondary | 0.903 | 0.321 | (0.740 1.101) | 0.904 | 0.563 | (0.642 1.273) | 0.882 | 0.315 | (0.691 1.126) | 1.074 | 0.786 | (0.641 1.799) |
| Tertiary | 0.430 | 0.000 | (0.272 0.682) | 0.871 | 0.681 | (0.451 1.682) | 0.449 | 0.001 | (0.276 0.730) | 1.071 | 0.853 | (0.517 2.219) |
| **Household Wealth index** | | | | | | | | | | | | |
| Poorest | 1 | | | 1 | | | 1 | | | 1 | | |
| Poorer | 0.802 | 0.003 | (0.694 0.927) | 0.851 | 0.202 | (0.664 1.090) | 0.8730 | 0.0780 | (0.750 1.015) | 0.8630 | 0.4330 | (0.597 1.247) |
| Middle | 0.729 | 0.000 | (0.624 0.852) | 0.689 | 0.020 | (0.504 0.943) | 0.7250 | 0.0000 | (0.606 0.867) | 0.5950 | 0.0250 | (0.378 0.937) |
| Richer | 0.646 | 0.000 | (0.531 0.785) | 0.770 | 1.177 | (0.526 1.126) | 0.7550 | 0.0120 | (0.606 0.940) | 1.0040 | 0.9860 | (0.632 1.595) |
| Richest | 0.438 | 0.000 | (0.334 0.575) | 0.591 | 1.041 | (0.357 0.9780) | 0.5120 | 0.0000 | (0.381 0.688) | 0.9360 | 0,830 | (0.508 1.722) |
| **Child's birth weight** | | | | | | | | | | | | |
| Very small | 1 | | | 1 | | | 1 | | | 1 | | |
| Small | 0.606 | 0.010 | (0.413 0.888) | 1.017 | 0.961 | (0.512 2.020) | 0.808 | 0.237 | (0.568 1.151) | 0.934 | 0.845 | (0.469 1.861) |
| Average or large | 0.360 | 0.000 | (0.253 0.511) | 0.613 | 0.128 | (0.326 1.151) | 0.426 | 0.000 | (0.309 0.587) | 0.585 | 0.094 | (0.312 1.096) |
| **Birth interval** | | | | | | | | | | | | |
| First birth | 1 | | | 1 | | | 1 | | | 1 | | |
| Less than 24 months | 1.178 | 0.060 | (0.993 1.396) | 1.032 | 0.840 | (0.756 1.410) | 1.224 | 0.054 | (0.997 1.504) | 1.012 | 0.958 | (0.647 1.584) |
| 25–47 months | 0.975 | 0.701 | (0.855 1.111) | 0.947 | 0.674 | (0.734 1.221) | 0.905 | 0.250 | (0.764 1.072) | 0.896 | 0.481 | (0.659 1.217) |
| 48+ months | 0.840 | 0.041 | (0.710 0.993) | 1.289 | 0.113 | (0.942 1.763) | 0.833 | 0.025 | (0.711 0.977) | 1.316 | 0.172 | (0.887 1.952) |
| **Mothers Employment Status** | | | | | | | | | | | | |
| Employed | 1 | | | 1 | | | 1 | | | 1 | | |
| Not employed | 0.992 | 0.885 | (0.884 1.112) | 1.115 | 0.249 | (0.926 1.342) | 1.001 | 0.979 | (0.882 1.138) | 0.831 | 0.230 | (0.614 1.125) |
| **Water source** | | | | | | | | | | | | |
| Improves | 1 | | | 1 | | | 1 | | | 1 | | |
| Non-improved | 1.100 | 0.089 | (0.985 1.229) | 1.045 | 0.698 | (0.838 1.303) | 1.114 | 0.127 | (0.970 1.280) | 0.928 | 0.668 | (0.658 1.309) |
| **Presence of diarrhea** | | | | | | | | | | | | |
| No | 1 | | | 1 | | | 1 | | | 1 | | |
| Yes, in the last 2 weeks | 1.046 | 0.480 | (0.923 1.184) | 1.110 | 0.393 | (0.873 1.411) | 0.906 | 0.217 | (0.774 1.060) | 1.143 | 0.510 | (0.767 1.704) |

**Table 4. Multivariate analysis of malnutrition indicators (Underweight and Overweight) for 2013/14 and 2018 DHS waves.**

| Background Characteristics | 2013–14 DHS | | | | | | 2018 DHS | | | | | |
| | Underweight | | | Overweight | | | Underweight | | | Overweight | | |
| | AOR | p-values | [95% CI] | AOR | p-values | [95% CI] | AOR | p-values | [95% CI] | AOR | p-values | [95% CI] |
| **Child's Age in months** | | | | | | | | | | | | |
| Less than 6 months | 1 | | | 1 | | | 1 | | | 1 | | |
| 6–8 | 2.080 | 0.001 | (1.377 3.142) | 0.740 | 0.096 | (0.519 1.054) | 1.350 | 0.205 | (0.849 2.147) | 0.366 | 0.000 | (0.232 0.579) |
| 9–11 | 3.030 | 0.000 | (2.001 4.587) | 0.699 | 0.078 | (0.469 1.041) | 1.626 | 0.032 | (1.044 2.532) | 0.322 | 0.000 | (0.190 0.546) |
| 12–17 | 2.342 | 0.000 | (1.633 3.358) | 0.382 | 0.000 | (0.270 0.539) | 2.001 | 0.000 | (1.389 2.882) | 0.317 | 0.000 | (0.205 0.492) |
| 18–23 | 3.324 | 0.000 | (2.351 4.699) | 0.368 | 0.000 | (0.258 0.526) | 2.173 | 0.000 | (1.488 3.173) | 0.330 | 0.000 | (0.223 0.488) |
| 24–35 | 3.257 | 0.000 | (2.377 4.61) | 0.217 | 0.000 | (0.156 0.302) | 1.765 | 0.000 | (1.291 2.413) | 0.240 | 0.000 | (0.163 0.352) |
| 36–47 | 2.803 | 0.000 | (2.057 3.820) | 0.207 | 0.000 | (0.149 0.288) | 1.933 | 0.000 | (1.369 2.729) | 0.209 | 0.000 | (0.142 0.309) |
| 48–59 | 2.736 | 0.000 | (2.003 3.737) | 0.166 | 0.000 | (0.117 0.234) | 1.512 | 0.015 | (1.085 2.106) | 0.136 | 0.000 | (0.089 0.208) |
| **Sex of Child** | | | | | | | | | | | | |
| Male | 1 | | | 1 | | | 1 | | | 1 | | |
| Female | 0.804 | 0.001 | (0.710 0.910) | 0.925 | 0.412 | (0.767 1.115) | 0.684 | 0.000 | (0.584 0.801) | 0.922 | 0.515 | (0.722 1.178) |
| **Residence** | | | | | | | | | | | | |
| Urban | 1 | | | 1 | | | 1 | | | 1 | | |
| Rural | 0.642 | 0.000 | (0.520 0.793) | 0.880 | 0.364 | (0.667 1.160) | 0.732 | 0.048 | (0.537 0.997) | 0.911 | 0.662 | (0.598 1.387) |
| **Education level of mother** | | | | | | | | | | | | |
| No education | 1 | | | 1 | | | 1 | | | 1 | | |
| Primary | 0.773 | 0.008 | (0.639 0.934) | 0.857 | 0.334 | (0.626 1.173) | 0.908 | 0.426 | (0.717 1.151) | 0.911 | 0.644 | (0.612 1.355) |
| Secondary | 0.719 | 0.007 | (0.566 0.914) | 0.793 | 0.227 | (0.543 1.156) | 0.776 | 0.077 | (0.585 1.028) | 0.865 | 0.558 | (0.532 1.406) |
| Tertiary | 0.360 | 0.001 | (0.193 0.670) | 1.023 | 0.948 | (0.521 2.010) | 0.847 | 0.607 | (0.448 1.599) | 1.061 | 0.892 | (0.452 2.488) |
| **Household Wealth index** | | | | | | | | | | | | |
| Poorest | 1 | | | 1 | | | 1 | | | 1 | | |
| Poorer | 0.749 | 0.000 | (0.638 0.879) | 0.908 | 0.502 | (0.686 1.202) | 0.866 | 0.150 | (0.711 1.054) | 0.775 | 0.119 | (0.561 1.068) |
| Middles | 0.563 | 0.000 | (0.467 0.679) | 1.233 | 0.153 | (0.925 1.645) | 0.592 | 0.000 | (0.451 0.778) | 0.792 | 0.250 | (0.532 1.179) |
| Richer | 0.444 | 0.000 | (0.338 0.582) | 1.053 | 0.787 | (0.723 1.535) | 0.646 | 0.007 | (0.470 0.888) | 0.921 | 0.766 | (0.536 1.584) |
| Richest | 0.325 | 0.000 | (0.231 0.457) | 1.215 | 0.462 | (0.723 2.039) | 0.557 | 0.012 | (0.352 0.879) | 0.828 | 0.570 | (0.432 1.587) |
| **Child's birth weight** | | | | | | | | | | | | |
| Very small | 1 | | | 1 | | | 1 | | | 1 | | |
| Small | 0.754 | 0.186 | (0.496 1.146) | 1.835 | 0.192 | (0.737 4.571) | 0.840 | 0.416 | (0.553 1.278) | 1.143 | 0.803 | (0.400 3.264) |
| Average or large | 0.338 | 0.000 | (0.227 0.503) | 2.321 | 0.047 | (1.012 5.325) | 0.344 | 0.000 | (0.233 0.510) | 1.706 | 0.247 | (0.690 4.222) |
| **Birth interval** | | | | | | | | | | | | |
| First birth | 1 | | | 1 | | | 1 | | | 1 | | |
| Less than 24 months | 1.352 | 0.01 | (1.076 1.699) | 1.115 | 0.556 | (0.776 1.603) | 1.319 | 0.025 | (1.035 1.680) | 1.069 | 0.767 | (0.686 1.667) |
| 25–47 months | 1.001 | 0.991 | (0.831 1.205) | 0.998 | 0.986 | (0.771 1.290) | 0.791 | 0.020 | (0.649 0.964) | 0.974 | 0.864 | (0.722 1.314) |
| 48+ months | 0.935 | 0.572 | (0.740 1.181) | 1.032 | 0.837 | (0.763 1.396) | 0.863 | 0.197 | (0.692 1.079) | 1.261 | 0.134 | (0.931 1.707) |
| **Mothers Employment Status** | | | | | | | | | | | | |
| Employed | 1 | | | 1 | | | 1 | | | 1 | | |
| Not employed | 1.041 | 0.599 | (0.896 1.209) | 1.060 | 0.592 | (0.856 1.312) | 0.804 | 0.015 | (0.675 0.978) | 0.810 | 0.103 | (0.629 1.043) |
| **Water source** | | | | | | | | | | | | |
| Improves | 1 | | | 1 | | | 1 | | | 1 | | |
| Nonimproved | 1.067 | 0.339 | (0.934 1.220) | 0.987 | 0.906 | (0.790 1.233) | 1.124 | 0.210 | (0.936 1.350) | 0.924 | 0.638 | (0.666 1.283) |
| **Presence of diarrhea** | | | | | | | | | | | | |
| No | 1 | | | 1 | | | 1 | | | 1 | | |
| Yes, in the last 2 weeks | 1.224 | 0.020 | (1.033 1.450) | 0.777 | 0.047 | (0.606 0.996) | 1.242 | 0.026 | (1.026 1.504) | 0.755 | 0.097 | (0.542 1.052) |

characteristics from the DHS data of 2013/4 and 2018. Although the prevalence of malnutrition decreased considerably between the survey years, the unequal burden of childhood malnutrition across the 2 survey periods was observed within provinces, with high prevalence reported in Luapula, Muchinga, and Northern provinces across most outcomes. Our results agree with results obtained from the Rural Agricultural Livelihoods Surveys (RALS) of 2012 and 2015 which concluded that Northern, Luapula, and Muchinga provinces had a high incidence of chronic poverty [22] thus increasing the risks of undernutrition. Our findings corroborate with studies done in LMIC within sub-Saharan Africa which found similar differences in the prevalence of malnutrition [1, 23, 24]. The information generated from this study showing the unequal burden of malnutrition is essential as it would help policymakers, clinicians, and nutrition fieldworkers plan regional tailored programs in responding to the different forms of malnutrition and address the underlying determinants of childhood malnutrition.

Our study has shown geographical disparities in the prevalence of stunting, wasting, underweight, and overweight suggesting the need for deliberate policies to reduce malnutrition. With the current population of 18 million, recent estimates project an annual population growth rate of 3% for Zambia [16]. Much of this population resides in rural areas (65% in 2010, 66% in 2013–14, and 57% in 2018) which are characterized by limited access to health and poor socio-economic conditions [16]. In addition, poverty remains high in rural areas in Zambia, with recent estimates indicating that 54% of Zambians are living on less than $1.90 a day [25]. Furthermore, changes in rainfall patterns and distribution due to climate change coupled with the rise in population, food price volatility, and cost of living increase the risks of food insecurity among rural families which may in turn influence the nutritional status of children in rural areas [26, 27].

Consistent with previous studies, the mother's educational status continues to be associated with childhood malnutrition, with lower odds across all indicators being observed among children whose mothers had tertiary education [28, 29]. Although education has been suggested to be linked to social status, our results suggest that maternal education can influence pathways such as income generation which may contribute to alleviating the burden caused by malnutrition. Furthermore, Iftikhar, Bari [30] suggested that educated mothers are better placed to adopt health-promoting behaviors, childhood feeding practices, and equitable sharing of resources within households thus significantly lowering childhood malnutrition. Furthermore, the acquisition of education plays a role in increasing responsiveness to health practices such as sanitation practices which are essential in reducing childhood risks to diarrheal [31, 32] and helminthic infections [33] which can lead to anaemia. Our study showed that sanitation in terms of having access to improved water sources did not influence malnutrition. The results obtained in our study are in agreement with earlier studies by Ukwuani and Suchindran [34] and Van de Poel, Hosseinpoor [35]. Despite this outcome, improved sanitation is vital in reducing the risks of diarrheal which was observed to increase the likelihood of being underweight; a result that has been reported in an earlier study conducted among children with malnutrition in Zambia [36].

Our results show that household wealth was significantly associated with stunting and underweight in children. We observed that children from poor households are more likely to experience poor outcomes compared to those from privileged households. Several studies have suggested a strong association between childhood malnutrition and the household's economic condition [37, 38]. Although our study like several other studies has observed that malnutrition is higher in perceived rural regions than urban regions due to household wealth differences, earlier studies showed that urban regions and cities have isolated clusters of communities living in severe poverty and deprivation compared to rural areas leading to risks of malnutrition among children from these households [39, 40]. On the other hand, high-

income households tend to spend more resources on nutrition and tend to have better access to healthcare, which in turn has a diverse impact on the health and nutritional status of the child [1, 41]. Primary school feeding programs have been implemented to reduce childhood malnutrition [42]. However, in areas where the rates of preschool and primary school dropouts are high, the impact of these programs may not be significant. Although these programs show gains in poverty reduction, there are possibilities of their implementations addressing symptoms of child malnutrition and not the determinants of childhood malnutrition. For instance, Roothaert, Mpogole [43] observed that school-based programs may lack food diversity and nutritive value and may not be enough for all students without contributions from parents.

While the prevalence of stunting and underweight appear to be more pronounced, our results suggest emerging patterns of child overweight in certain regions such as Lusaka and Northern provinces. Amidst the rising cases of overweight/obesity in children, more studies in LMICs tend to focus more on the contributory role of poverty and malnutrition, without considering other important contextual factors [23]. The development of policies and programs that address food security, maternal education, and socio-economic inequalities will be essential in mitigating the long-term impacts of childhood malnutrition on a child's intellectual development. Our findings suggest the need for policies that will address the proximate determinants of childhood malnutrition. There is a need to strength policies, and programs targeting specific populations such as children from poor families and communities to achieve pro-parity health outcomes. With the growing population of Zambia, is a pressing need for policies to alleviate childhood malnutrition; designed through community involvement and consultation to enhance participation, especially in the northern regions of the country.

## Study strengths and study limitations

A major strength of our study is the utility of nationally representative survey data that allowed us to examine the prevalence of childhood malnutrition in 2013/4 and 2018. The study was limited to the explanatory factors for which data was collected and reported in DHS 2013 and DHS 2018. Thus, data on many potential correlates/other risk factors such as access to the health facility, adolescent pregnancy, comorbidities, and NCDs were not included in the analysis. The available data was measured during the survey period and may have changed.

## Conclusions

Health equity is a critical component for achieving Sustainable Development Goals (SDG 3). Furthermore, the WHO has identified geographical disparities and inequalities in inequitable implementation of intervention programs. Our findings contribute to the understanding of the etiology and epidemiology of child malnutrition in a bid to reduce persistent inequalities in childhood malnutrition. In particular, the findings highlight regional variations and key factors driving the trends in childhood malnutrition between the two survey periods. Our study suggests that reducing malnutrition in Zambia will require specific regional tailored interventions in the identified provinces. Furthermore, in-depth studies on other risk factors related to childhood malnutrition should also be conducted. The country should develop a core set of nutritional indicators to be monitored rapidly/frequently across provinces and on all the nutritional programs.

## Acknowledgments

Thanks go to the Demographic and Health Survey Program for the approval to use the 2013–14 and 2018 Zambia Demographic and Health Survey datasets for analysis.

## Author Contributions

**Conceptualization:** Million Phiri, David Mulemena, Chester Kalinda, Julius Nyerere Odhiambo.

**Data curation:** Million Phiri, David Mulemena, Chester Kalinda, Julius Nyerere Odhiambo.

**Formal analysis:** Million Phiri, Chester Kalinda, Julius Nyerere Odhiambo.

**Methodology:** Million Phiri, Julius Nyerere Odhiambo.

**Project administration:** Chester Kalinda.

**Software:** Million Phiri, David Mulemena, Chester Kalinda.

**Supervision:** Chester Kalinda.

**Validation:** Million Phiri, David Mulemena, Julius Nyerere Odhiambo.

**Visualization:** Chester Kalinda.

**Writing – original draft:** Chester Kalinda, Julius Nyerere Odhiambo.

**Writing – review & editing:** Million Phiri, Chester Kalinda, Julius Nyerere Odhiambo.

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
