## [Decision Letter · Decision Letter 0]

2 Mar 2022

PONE-D-21-18211Contextual Factors and Spatial Trends of Childhood Malnutrition in Zambia

PLOS ONE

Dear Dr. Odhiambo,

Thank you for submitting your manuscript to PLOS ONE. After careful consideration, we feel that it has merit but does not fully meet PLOS ONE’s publication criteria as it currently stands. Therefore, we invite you to submit a revised version of the manuscript that addresses the points raised during the review process.

The manuscript has been evaluated by two reviewers, and their comments are available below. The reviewers have raised a number of concerns. They request improvements to the reporting of methodological aspects of the study, for example, regarding the exclusion criteria and more information on how the data collection was completed. The reviewers also note concerns about the statistical analyses presented. Could you please carefully revise the manuscript to address all comments raised?

We look forward to receiving your revised manuscript.

Kind regards,

Elisa Panada

Associate Editor

PLOS ONE

Journal Requirements:

2. We note that Figure 2 in your submission contain map images which may be copyrighted. All PLOS content is published under the Creative Commons Attribution License (CC BY 4.0), which means that the manuscript, images, and Supporting Information files will be freely available online, and any third party is permitted to access, download, copy, distribute, and use these materials in any way, even commercially, with proper attribution. For these reasons, we cannot publish previously copyrighted maps or satellite images created using proprietary data, such as Google software (Google Maps, Street View, and Earth). For more information, see our copyright guidelines: http://journals.plos.org/plosone/s/licenses-and-copyright.

3. Please include your tables as part of your main manuscript and remove the individual files. Please note that supplementary tables (should remain/ be uploaded) as separate "supporting information" files

4. Thank you for including your ethics statement: "The 2013-14 and 2018 Zambia Demographic and Health Survey protocols for survey methodology and biomarker measurements were approved by institutional review boards (IRBs) at ICF and the Tropical Diseases Research Centre (TDRC) in Zambia. Both IRBs approved the protocols before the commencement of data collection activities. Informed consent was obtained by enumerators before the commencement of interviews in all selected households. Permission to use DHS data for this study was sought from ICF international [28]." 

a) Please provide additional details regarding participant consent. In the ethics statement in the Methods and online submission information, please ensure that you have specified what type you obtained (for instance, written or verbal, and if verbal, how it was documented and witnessed). If your study included minors, state whether you obtained consent from parents or guardians. If the need for consent was waived by the ethics committee, please include this information.

Reviewers' comments:

Reviewer's Responses to Questions

**Comments to the Author**

1. Is the manuscript technically sound, and do the data support the conclusions?

Reviewer #1: Yes

Reviewer #2: Partly

2. Has the statistical analysis been performed appropriately and rigorously? 

Reviewer #1: Yes

Reviewer #2: Yes

3. Have the authors made all data underlying the findings in their manuscript fully available?

Reviewer #1: No

Reviewer #2: Yes

4. Is the manuscript presented in an intelligible fashion and written in standard English?

Reviewer #1: Yes

Reviewer #2: Yes

5. Review Comments to the Author

Reviewer #1: The manuscript is written well but with unknown reason I did not access the tables. However, the presented information is enough to forward my concerns on the paper.

1. As the abstract is the stand alone summary of the whole work; information like the total number of participants involved in the analysis must be included. It might be editorial problem use multivariate vs multivariable appropriately; in the paper it is used interchangeably; which is not right.

2. Though the introduction section address key points; it needs further modification; especially the figurative expressions of each malnutrition from the global to local context must be incorporated. If possible also add the global and the national trend from the previous reports. In addition, as mentioned in the title, "contextual factors", factors associated with each malnutrition type must be addressed in the introduction section.

3. In the result section, particularly the bivariate and the multivariable analysis section; most of the measure of effects are below 1 (null value for odds ratio); this is due to the selection of the reference category; why you did that? In addition, though the measure of effect is put with its 95% CI; its interpretation is a bit confusing. Just an example let me raise this one: "In 2014, children aged 48-59 months were 0.17(95% CI: 0.12–0.23) times less likely to be overweight". First, better to change the reference category with the smallest percentage or you have to interpret as In 2014, children aged 48-59 months were 83% (AOR: 0.17; 95% CI: 0.12–0.23) less likely to be overweight".

4. In the discussion; section from lines 240-243, is it the justification given for the similarities/discrepancies of your findings and the previous findings or recommendation? I think this is recommendation and take it this to the appropriate place and put the justification for the similarities/discrepancies of the findings.

Reviewer #2: Overall – a well written manuscript reporting on an analysis from the DHS that looks at geographical differences in child malnutrition and sociodemographic predictors within Zambia; however, the manuscript could be strengthened by increasing the focus on these geographic variations and highlighting the importance of the sociodemographic characteristics of provinces within Zambia in relation with the outcomes.

Specific recommendations follow:

Abstract –

The justification for the study is that there are within country variations that are important beyond the national estimates and the methods describe assessment of within province variations, however the results only provide changes in national estimates from 2013 to 2018. The results need to be updated to support the rest of the abstract, including the conclusions.

I do not think that there was a decrease in overweight based on the estimates provided. I would say, it remained about the same.

Introduction –

37- It would be good to clarify what is meant by malnutrition, which is normally used for both under and over nutrition. While undernutrition does account for childhood mortality and impairs development, the role of overweight in childhood is not as clear. You could also mention the long-lasting consequences for chronic diseases.

68 – Please clarify what is meant by “a relative shift in the contribution of socio-economic determinants, communicable and non-communicable diseases”. Also provide a citation for this statement.

Methods –

84 – there seems to be a typo “of women aged of selected households”.

84- please clarify the concept of biological children, was this asked during a screening? Were adopted children excluded?

86- Was child weight and height measured or reported by the mothers? Since the study is in children, I do not understand what is meant by women participants. Please clarify this in the study design or sample selection. Also, please distinguish between the inclusion criteria for the DHS and the inclusion criteria for this analysis.

99- The statement about malnutrition and this study including over- and under- nutrition is unclear. Please rephrase to clarify.

99-101 Perhaps this information is better suited in the introduction and/or discussion rather than in the methods.

Statistical analysis – is the survey designed to be representative at the province level? If so, please state it in the study description.

Please add a section describing the covariates for the models. The case for studying these associations should be made in the introduction.

Results

123-125 – it sounds like the variables were included in the model based on their known importance as predictors. Perhaps that is a simpler way to state this (and in general a better one than basing it on significance). (There is a typo – “interested variables” should be variables of interest).

169 – what is meant by uncertainty? This is not defined before in the methods.

The analysis of malnutrition by province needs to be expanded to describe highest and lowest prevalence of under- and over- nutrition, as well as greatest changes over time.

Additional models correlating geographic or province sociodemographic characteristics to malnutrition outcomes would greatly increase the value of this paper. Please consider including these additional analysis which can be performed either by multilevel modeling with a province level or simple spatial modeling using the ARCGis

Conclusion

Even though the introduction emphasizes the relevance of geographic differences, the conclusion just presents these as an afterthought and concentrates on the associations between sociodemographic characteristics and malnutrition. There is a missed opportunity to integrate these two in a discussion about the importance of geographical variation and changes over time, and what the role of the characteristics of the different provinces means. I recommend re-writing the results and discussion, and potentially adding more analysis to fully address the geographic differences, rather than associations that are in general expected for child malnutrition and do not inform policies or interventions.

6. PLOS authors have the option to publish the peer review history of their article (what does this mean?). If published, this will include your full peer review and any attached files.

Reviewer #1: No

Reviewer #2: No

---

## [Decision Letter · Decision Letter 1]

12 Jul 2022

PONE-D-21-18211R1

Contextual factors and spatial trends of childhood malnutrition in Zambia

PLOS ONE

Dear Dr. Odhiambo,

Thank you for submitting your manuscript to PLOS ONE. After careful consideration, we feel that it has merit but does not fully meet PLOS ONE’s publication criteria as it currently stands. As context, I served as reviewer 2 for the initial submission of the manuscript and then agreed to serve as guest editor for this resubmission. This led me to the decision to invite an additional reviewer to account for any potential bias. While previous comments have been addressed, the new reviewer found that major changes needed to be done before the manuscript is suitable for publication. Therefore, we invite you to submit a revised version of the manuscript that addresses the points raised during the review process.

We look forward to receiving your revised manuscript.

Kind regards,

Ines Gonzalez Casanova

Guest Editor

PLOS ONE

Reviewers' comments:

Reviewer's Responses to Questions

**Comments to the Author**

1. If the authors have adequately addressed your comments raised in a previous round of review and you feel that this manuscript is now acceptable for publication, you may indicate that here to bypass the “Comments to the Author” section, enter your conflict of interest statement in the “Confidential to Editor” section, and submit your "Accept" recommendation.

Reviewer #1: All comments have been addressed

Reviewer #3: All comments have been addressed

2. Is the manuscript technically sound, and do the data support the conclusions?

Reviewer #1: Yes

Reviewer #3: No

3. Has the statistical analysis been performed appropriately and rigorously? 

Reviewer #1: Yes

Reviewer #3: I Don't Know

4. Have the authors made all data underlying the findings in their manuscript fully available?

Reviewer #1: Yes

Reviewer #3: No

5. Is the manuscript presented in an intelligible fashion and written in standard English?

Reviewer #1: Yes

Reviewer #3: Yes

6. Review Comments to the Author

Reviewer #1: (No Response)

Reviewer #3: I was excited to be considered to review the manuscript titled:

Contextual factors and spatial trends of childhood malnutrition in Zambia

The manuscript attempts to write about malnutrition in Zambia using DHA data from two different surveys.

Mi biggest comment is that the manuscript has an flaw that needs to be addressed: there is not a clear research question, or questions, or hypotheses.

The main outcome is malnutrition, but relative to what? Is it changes? Or is it identifying factors? Or identifying factors that changed over time? ?

Based on the hypothesis/research questions, the authors have an amazing opportunity to use the scientific method to respond to the main questions they are asking. Right now, it reads as a report of the two DHA surveys, which I don’t think it is suitable for publication.

ABSTRACT:

The abstract has the following conclusion: “The study points to key sub-populations at greater risk and provinces where malnutrition was prevalent in Zambia”. The results included in the abstract explain reductions in the odds ratios of stunting, wasting, underweight and overweight. The conclusion is not related to the results included in the abstract. I suggest the authors modify the abstract in such way that the most important results from the manuscript are listed, and the conclusion is based on the most important findings of the study.

OVERALL: undernutrition and malnutrition are two different concepts. The authors should consider revising the manuscript and use the appropriate term when needed: https://www.who.int/news-room/fact-sheets/detail/malnutrition

In the introduction, the authors consistently are using “malnutrition” when referring to “undernutrition”.

What are the associated issues with overweight and obesity during childhood? And double burden?

I suggest to work a bit more on working more on the “ problem” that under and over nutrition has, especially within the context of LMIC, Africa and Zambia.

Introduction:

The Lancet global series has new references on unde- and over- rnutrition in LMIC. Consider including some more updated references on worldwide undernutrition:

See here: https://www.thelancet.com/series/maternal-child-undernutrition-progress

Line 46 needs a reference.

Consider rewording 76-78

METHODS:

Table 1 should be supplementary material.

Undernourishment,. Consider changing to undernutrition

Line 148 is malnutrition? Or undernutrition? (is it also overnutrition?)

The tables are very large. Consider including the most important variales.

Where is table 3?

Tables 2 are showing stratified ages, however only a p-value is shown when comparing the age+ stunting, wasting a the age used as a contibnuous variable? It is not clear what the P-values are comparing? Example: is it male vs female in stunted? Is it stunting for males and females? How did the authors analize the age variable?

Outcome measures: theauthors refer that the outcome is childgood malnutrition), but relative to what?

Is it time changes? Is it factors?

Based on the main research question of this manuscript, then the authors should consuider using generalized linear mixed models, as there are two potential clsuters of data that should be accounted for as random effects. (Although I am not a biostatistician, so I would defer to other reviewers with more knowledge on this topic).

1) Years

2) Provinces.

7. PLOS authors have the option to publish the peer review history of their article (what does this mean?). If published, this will include your full peer review and any attached files.

Reviewer #1: No

Reviewer #3: No

---

## [Author Response · Author response to Decision Letter 1]

16 Aug 2022

The manuscript attempts to write about malnutrition in Zambia using DHA data from two different surveys.

Mi biggest comment is that the manuscript has an flaw that needs to be addressed: there is not a clear research question, or questions, or hypotheses.

The main outcome is malnutrition, but relative to what? Is it changes? Or is it identifying factors? Or identifying factors that changed over time? ? 

In this paper we intended to look at the individual socioeconomic, demographic and contextual risk factors associated with malnutrition between two DHS survey periods.

As defined by the WHO, childhood malnutrition, in all its forms, includes undernutrition (wasting, stunting, underweight), inadequate vitamins or minerals, overweight, obesity, and resulting diet-related noncommunicable diseases. More emphasis has been added in the manuscript. Please see lines 43 -45 and 97 - 104

Based on the hypothesis/research questions, the authors have an amazing opportunity to use the scientific method to respond to the main questions they are asking. Right now, it reads as a report of the two DHA surveys, which I don’t think it is suitable for publication. Our study sought to highlights the geographical disparity of childhood malnutrition using the 2013 and 2018 ZDHS data. 

Our hypothesis is that childhood malnutrition patterns change in time and space.

ABSTRACT:

The abstract has the following conclusion: “The study points to key sub-populations at greater risk and provinces where malnutrition was prevalent in Zambia”. The results included in the abstract explain reductions in the odds ratios of stunting, wasting, underweight and overweight. The conclusion is not related to the results included in the abstract. I suggest the authors modify the abstract in such way that the most important results from the manuscript are listed, and the conclusion is based on the most important findings of the study. Agreed.

Our study highlights the overall trends across all the provinces and the heterogeneity between the two periods.

We have reworded our abstract to enhance its clarity.

OVERALL: undernutrition and malnutrition are two different concepts. The authors should consider revising the manuscript and use the appropriate term when needed: https://www.who.int/news-room/fact-sheets/detail/malnutrition

In the introduction, the authors consistently are using “malnutrition” when referring to “undernutrition”.

What are the associated issues with overweight and obesity during childhood? And double burden?

I suggest to work a bit more on working more on the “ problem” that under and over nutrition has, especially within the context of LMIC, Africa and Zambia. According to the World Health Organization, malnutrition refers to “deficiencies, excesses or imbalances in a person’s intake of energy and/or nutrients.” 

This can result in either undernutrition or overweight.

The introduction has been refined and additional references included.

Line 76 -78 has also been reworded. Please see lines 75 - 79

Introduction:

The Lancet global series has new references on unde- and over- rnutrition in LMIC. Consider including some more updated references on worldwide undernutrition:

See here: https://www.thelancet.com/series/maternal-child-undernutrition-progress

Line 46 needs a reference.

Consider rewording 76-78 

METHODS:

Table 1 should be supplementary material.

Undernourishment,. 

Consider changing to undernutrition Agreed.

Table 1 has been removed. 

Line 148 is malnutrition? Or undernutrition? (is it also overnutrition?) According to the World Health Organization, malnutrition refers to “deficiencies, excesses or imbalances in a person’s intake of energy and/or nutrients.” 

This can result in either undernutrition or overweight.

The tables are very large. Consider including the most important variales.

Where is table 3? 

Table 3 is clearly annotated.

Tables 2 are showing stratified ages, however only a p-value is shown when comparing the age+ stunting, wasting a the age used as a contibnuous variable? It is not clear what the P-values are comparing? Example: is it male vs female in stunted? Is it stunting for males and females? How did the authors analize the age variable? The association between categorical variables were analysed the chi-square test.

The p-values highlights if the association was significant. 

Based on the main research question of this manuscript, then the authors should consuider using generalized linear mixed models, as there are two potential clsuters of data that should be accounted for as random effects. (Although I am not a biostatistician, so I would defer to other reviewers with more knowledge on this topic).

1) Years

2) Provinces. We used a logistic regression model since our dependent variable (malnutrion) was binary.

---

## [Editor Report · Decision Letter 2]

21 Sep 2022

PONE-D-21-18211R2Contextual factors and spatial trends of childhood malnutrition in ZambiaPLOS ONE

Dear Dr. Odhiambo,

Thank you for submitting your manuscript to PLOS ONE. After careful consideration, we feel that it has merit but does not fully meet PLOS ONE’s publication criteria as it currently stands. Therefore, we invite you to submit a revised version of the manuscript that addresses the points raised during the review process.

ACADEMIC EDITOR: While the manuscript is almost ready for publication, the latest version still did not fully address  the reviewer's comments. Minor edits are necessary so that the manuscript can be accepted. Please address the following comments:

Abstract: The results still do not reflect the regional differences that are described in the background, methods, and conclusion. You need to describe at least some evidence of heterogeneity in the results.

To address concerns related to the use of different terminology to describe undernutrition or malnutrition, I suggest that you add a footnote explaining how you are using each term in each context and that you are consistent with the use across.

Please limit the tables to key variables that illustrate this study and the results. This will address the reviewer's comment that states that this looks more like a report than an analysis.

Add this information to table legends (include comparison groups, dependent variable, independent variable, co-variates, how variables were analyzed, etc).:

*Tables 2 are showing stratified ages, however only a p-value is shown when comparing the age+ stunting, wasting a the age used as a contibnuous variable? It is not clear what the P-values are comparing? Example: is it male vs female in stunted? Is it stunting for males and females? How did the authors analize the age variable?The association between categorical variables were analysed the chi-square test. The p-values highlights if the association was significant. Based on the main research question of this manuscript, then the authors should consuider using generalized linear mixed models, as there are two potential clsuters of data that should be accounted for as random effects. (Although I am not a biostatistician, so I would defer to other reviewers with more knowledge on this topic). 1) Years 2) Provinces.We used a logistic regression model since our dependent variable (malnutrion) was binary.*

Make sure all tables and figures have descriptive legends and can be interpreted on their own. 

A rebuttal letter that responds to each point raised by the academic editor. You should upload this letter as a separate file labeled 'Response to Reviewers'.A marked-up copy of your manuscript that highlights changes made to the original version. You should upload this as a separate file labeled 'Revised Manuscript with Track Changes'.An unmarked version of your revised paper without tracked changes. You should upload this as a separate file labeled 'Manuscript'.If applicable, we recommend that you deposit your laboratory protocols in protocols.io to enhance the reproducibility of your results. Protocols.io assigns your protocol its own identifier (DOI) so that it can be cited independently in the future. For instructions see: https://journals.plos.org/plosone/s/submission-guidelines#loc-laboratory-protocols. Additionally, PLOS ONE offers an option for publishing peer-reviewed Lab Protocol articles, which describe protocols hosted on protocols.io. Read more information on sharing protocols at https://plos.org/protocols?utm_medium=editorial-email&utm_source=authorletters&utm_campaign=protocols.

We look forward to receiving your revised manuscript.

Kind regards,

Inés González-Casanova

Guest Editor

PLOS ONE

Journal Requirements:

Additional Editor Comments:

Abstract: The results still do not reflect the regional differences that are described in the background, methods, and conclusion. You need to describe at least some evidence of heterogeneity in the results.

To address concerns related to the use of different terminology to describe undernutrition or malnutrition, I suggest that you add a footnote explaining how you are using each term in each context and that you are consistent with the use across.

Please limit the tables to key variables that illustrate this study and the results. This will address the reviewer comment that states that this looks more like a report than an analysis.

Add this information to table legends (include comparison groups, dependent variable, independent variable, co-variates, how variables were analyzed, etc).:

Tables 2 are showing stratified ages, however only a p-value is shown when comparing the age+ stunting, wasting a the age used as a contibnuous variable? It is not clear what the P-values are comparing? Example: is it male vs female in stunted? Is it stunting for males and females? How did the authors analize the age variable?The association between categorical variables were analysed the chi-square test. The p-values highlights if the association was significant. Based on the main research question of this manuscript, then the authors should consuider using generalized linear mixed models, as there are two potential clsuters of data that should be accounted for as random effects. (Although I am not a biostatistician, so I would defer to other reviewers with more knowledge on this topic). 1) Years 2) Provinces.We used a logistic regression model since our dependent variable (malnutrion) was binary.

---

## [Author Response · Author response to Decision Letter 2]

4 Oct 2022

R1

Abstract: The results still do not reflect the regional differences that are described in the background, methods, and conclusion. You need to describe at least some evidence of heterogeneity in the results.

AR1:Agreed.

We have added more information on the regional disparities. Please see lines 35 - 38

R2

To address concerns related to the use of different terminology to describe undernutrition or malnutrition, I suggest that you add a footnote explaining how you are using each term in each context and that you are consistent with the use across.

AR2: Agreed

The WHO and Zambian standards have been used to define the anthropometric indicators of nutritional status i.e. stunting, wasting, underweight and overweight. Please see lines 98 - 106

R3

Please limit the tables to key variables that illustrate this study and the results. This will address the reviewer's comment that states that this looks more like a report than an analysis.

AR3: Agreed. 

Please see Table 1 and 2.

R4

Add this information to table legends (include comparison groups, dependent variable, independent variable, co-variates, how variables were analyzed, etc).

AR4: Agreed. Please see Table 1 and 2.

R5

Tables 2 are showing stratified ages, however only a p-value is shown when comparing the age+ stunting, wasting a the age used as a contibnuous variable? It is not clear what the P-values are comparing? Example: is it male vs female in stunted? Is it stunting for males and females? How did the authors analize the age variable?The association between categorical variables were analysed the chi-square test. The p-values highlights if the association was significant. Based on the main research question of this manuscript, then the authors should consuider using generalized linear mixed models, as there are two potential clsuters of data that should be accounted for as random effects. (Although I am not a biostatistician, so I would defer to other reviewers with more knowledge on this topic). 1) Years 2) Provinces.We used a logistic regression model since our dependent variable (malnutrion) was binary.

AR5: 

This seems to be a verbatim repetition of the initial reviewer comments and our response.

---

## [Editor Report · Decision Letter 3]

19 Oct 2022

Contextual factors and spatial trends of childhood malnutrition in Zambia

PONE-D-21-18211R3

Dear Dr. Odhiambo,

We’re pleased to inform you that your manuscript has been judged scientifically suitable for publication and will be formally accepted for publication once it meets all outstanding technical requirements.

Kind regards,

Inés González-Casanova

Guest Editor

PLOS ONE

Additional Editor Comments (optional):

Please explain what is meant by uncertainty in the abstract. Perhaps a better term is variability within the province? That result is not clear and should be better explained.
---

## [Editor Report · Acceptance letter]

25 Oct 2022

PONE-D-21-18211R3 

Contextual factors and spatial trends of childhood malnutrition in Zambia 

Dear Dr. Odhiambo:

I'm pleased to inform you that your manuscript has been deemed suitable for publication in PLOS ONE. Congratulations! Your manuscript is now with our production department. 

Kind regards, 

on behalf of

Dr. Inés González-Casanova 

Guest Editor

PLOS ONE